# Britholite Group Minerals from REE-Rich Lithologies of Keivy Alkali Granite—Nepheline Syenite Complex, Kola Peninsula, NW Russia

**Dmitry Zozulya** [1,*], **Lyudmila Lyalina** [1], **Ray Macdonald** [2,3], **Bogusław Bagiński** [2], **Yevgeny Savchenko** [1] **and Petras Jokubauskas** [2]

1    Geological Institute, Kola Science Centre, 14 Fersman Str, 184209 Apatity, Russia; lialina@geoksc.apatity.ru (L.L.); evsav@geoksc.apatity.ru (Y.S.)
2    Institute of Geochemistry, Mineralogy and Petrology, University of Warsaw, al. Żwirki i Wigury 93, 02089 Warsaw, Poland; r.macdonald@lancaster.ac.uk (R.M.); b.baginski1@uw.edu.pl (B.B.); p.jokubauskas@uw.edu.pl (P.J.)
3    Environment Centre, Lancaster University, Lancaster LA1 4YQ, UK
*    Correspondence: zozulya@geoksc.apatity.ru; Tel.: +7-81555-79742

**Abstract:** The Keivy alkali granite-nepheline syenite complex, Kola Peninsula, NW Russia, contains numerous associated Zr-REE-Y-Nb occurrences and deposits, formed by a complex sequence of magmatic, late-magmatic, and post-magmatic (including pegmatitic, hydrothermal, and metasomatic) processes. The REE-rich lithologies have abundant (some of economic importance) and diverse britholite group minerals. The REE and actinides distribution in host rocks indicates that the emanating fluids were alkaline, with significant amounts of F and $CO_2$. From chemical studies (REE and F variations) of the britholites the possible fluid compositions in different lithologies are proposed. Fluorbritholite-(Y) and britholite-(Y) from products of alkali granite (mineralized granite, pegmatite, quartzolite) formed under relatively high F activity in fluids with low $CO_2/H_2O$ ratio. The highest F and moderate $CO_2$ contents are characteristic of fluid from a mineralized nepheline syenite, resulting in crystallization of fluorbritholite-(Ce). Britholite group minerals (mainly fluorcalciobritholite and 'calciobritholite') from a nepheline syenite pegmatite formed from a fluid with composition changing from low F and high $CO_2$ to moderate F and $CO_2$. An extremely high F content is revealed for metasomatizing fluids emanating from alkali granitic magma and which affected the basic country rocks. The dominant substitution scheme for Keivy britholites is $REE^{3+} + Si^{4+} = Ca^{2+} + P^{5+}$, showing the full range of 'britholite' and 'calciobritholite' compositions up to theoretical apatite.

**Keywords:** britholite group minerals; alkali granite; nepheline syenite; pegmatite; quartzolite; metasomatite; fluid composition; Keivy; Kola Peninsula; Russia

## 1. Introduction

The britholite group contains relatively common accessory minerals that are considered to be among the main carriers of the rare-earth elements (REE) in certain igneous rocks. The minerals have been identified in felsic magmatic rocks, and in metasomatites and pegmatites associated with subalkali and alkali granites, alkali and nepheline syenites, alkali volcanics, and carbonatites [1–16]. It has been suggested that britholites in most occurrences crystallized at the late- to post-magmatic (pegmatite and hydrothermal) stages. Yttrium-rich britholites are typical of granites and their derivatives, whilst the cerium-dominant species occur in nepheline syenites and carbonatites. In addition, in some complex rare-metal deposits related to felsic and intermediate alkaline rocks (Mushugui [14]; Eden Lake [2]; West Keivy [17]; Azovskoe [18]; Rodeo de Los Molles [19,20]; Sakharjok [21]; Misery [22]; Weishan

and Maoniuping [23]; Pilanesberg and Ilimaussaq [24]), britholite is not a traditional ore mineral representing an economically important source of REE.

The britholite group belongs to the apatite supergroup and currently includes (with the exception of the tritomite species) britholite-(Ce), britholite-(Y), fluorbritholite-(Ce), fluorbritholite-(Y), and fluorcalciobritholite [25]. In the literature, the potential phases "calciobritholite" [8,16] and IMA valid "britholite-(La)" [15] have also been mentioned. The group includes minerals with the general chemical formula $^{IX}M1_2{}^{VII}M2_3(^{IV}TO_4)_3X$ where M = $Ca^{2-}$, $Ce^{3-}$, $La^{3-}$, $Y^{3-}$; T = $P^{5-}$, $Si^{4-}$; X = $F^-$, $(OH)^-$, $Cl^-$. The chemical compositions of these minerals are highly variable with different REE/Ca, P/Si and F/OH ratios depending mainly on the chemistry of the environment of formation and reflect complex isomorphic solid solutions among the britholite end-members. Several species may crystallize in a single rock due to different genetic processes (e.g., in UK Paleogene granites [8], the Stupne granite, Slovakia [16], and the Sakharjok nepheline syenite, Kola Peninsula, NW Russia [21,26].

In this paper, we provide textural and compositional data for britholite-group minerals (BGM) forming abundant REE-bearing minerals in lithologies of different genetic types from the Late Archean Keivy alkali granite – nepheline syenite complex in the Kola Peninsula, NW Russia. The lithologies are mainly mineralized granites and syenites, as well as pegmatites, quartzolites, and metasomatic rocks. A specific aim is to determine the nature and composition of the late- to postmagmatic fluids and how they are reflected in BGM composition.

An economically important REE-Zr deposit has been explored in the Sakharjok nepheline syenite massif from Keivy with BGM as the main REE ore minerals [21,27]. An effective (low-cost, high recovery of individual REE) mineral processing scheme based on hydrometallurgical processes has been developed for REE extraction from the Sakharjok BGM concentrate [28].

## 2. Geological Background and REE-Rich Occurrences

The Keivy alkaline province consists of 2.67–2.65 Ga aegirine-arfvedsonite granites (six sheet-like massifs a few hundred meters thick and with a total exposure area of ca. 2500 km$^2$). The largest massifs are West Keivy (1200 km$^2$), Ponoj (700 km$^2$) and White Tundra (120 km$^2$). Aegirine-augite-lepidomelane-ferrohastingsite syenogranites occur at the margins of some massifs. The granites intrude the TTG basement of the Central Kola terrane (NE Fennoscandian shield) and acid-intermediate metavolcanics of the Lebyazha formation of the Keivy complex (Figure 1). Lepidomelane-ferrohastingsite syenite and lepidomelane-aegirine nepheline syenite dykes outcrop within the West Keivy massif. The Keivy alkali granite-nepheline syenite complex is one of the earliest known A-type suites on Earth [29,30]. The voluminous coeval (ca. 2660 Ma [30]) gabbro-anorthosites are spatially confined to the Keivy alkali granites.

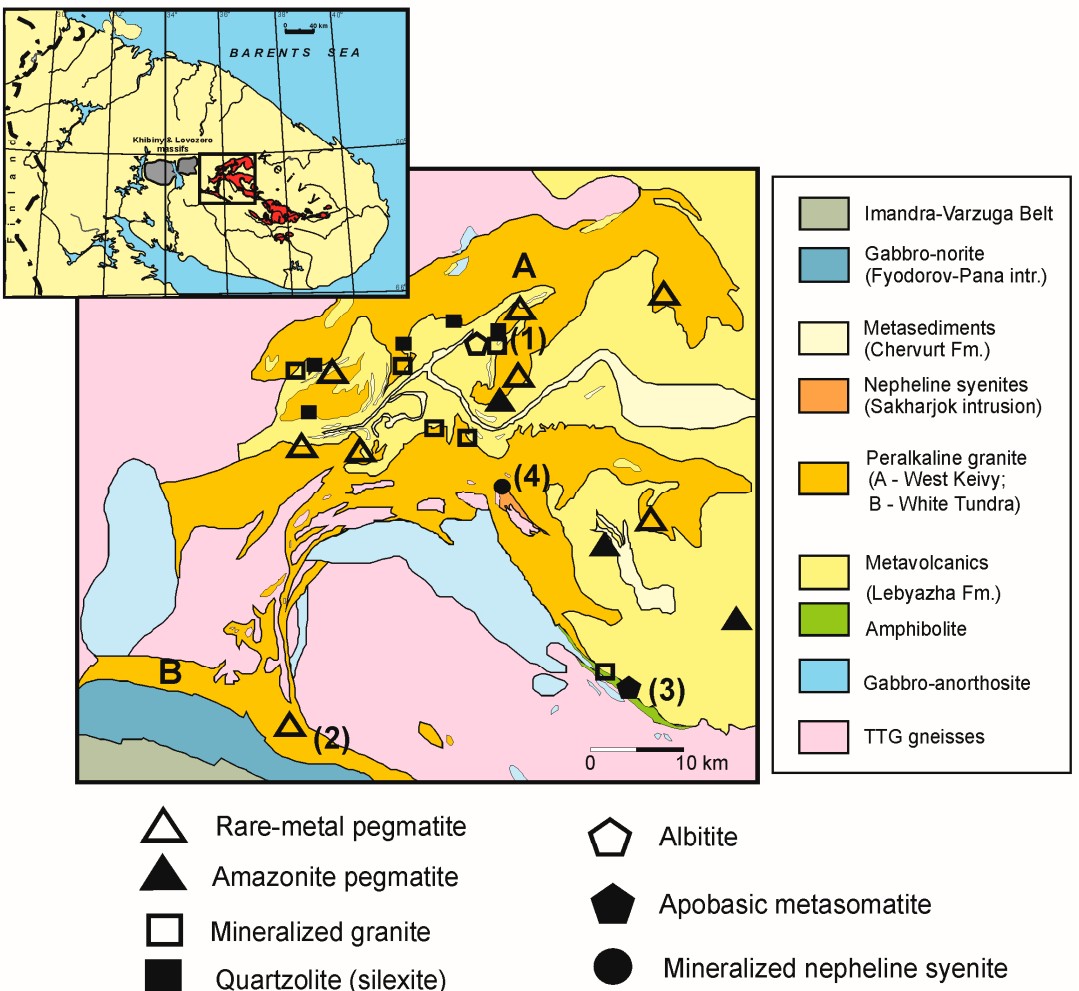

**Figure 1.** Simplified geological map of the western part of the Keivy alkaline province, Kola Peninsula, NW Russia, showing the location of Zr-REE-Y-Nb ore occurrences and deposits associated with different lithologies. Numbered occurrences are studied in this paper (1—Rova (mineralized alkali granite, quartzolite); 2—White Tundra pegmatite; 3—El'ozero quartz-epidote (apobasic) metasomatite; 4—Sakharjok mineralized nepheline syenite, pegmatite and micaceous metasomatite). The insert shows the location and structure of the Keivy alkali granite complex in the Kola Peninsula (boxed area).

The rocks of the Keivy province are extremely enriched in Zr (300–5000 ppm), Y (40–500 ppm), Nb (20–600 ppm), REE (100–1000 times chondrite), which has been explained by an enriched mantle source for the primary melts and extreme fractionation processes [27,29,30]. Numerous Zr-Y-REE-Nb ore occurrences and deposits are associated with different lithologies and were formed by various petrogenetic (late- and post-magmatic) processes (Figure 1). A summary of common and accessory, primary, and secondary rare-metal minerals, as well as whole-rock content of REE, Zr, Nb, Y, Th, and U, are given in Table 1.

**Table 1.** Major, minor and accessory REE-Nb-Zr-Th-U-Be minerals and whole-rock content of REE, Nb, Zr, Th, and U in rare-metal rich lithologies related to the Keivy alkali granite-nepheline syenite complex.

| Mineralized Granite | Alkali Granite Pegmatite | Quartzolite | Apobasic Quartz-Epidote Metasomatite | Mineralized Nepheline Syenite | Nepheline Syenite Pegmatite |
|---|---|---|---|---|---|
| **zircon**, **fergusonite-(Y)**, **chevkinite-(Ce)**, *bastnäsite-(Ce)*, monazite-(Ce), britholite-(Y), allanite-(Ce), thorite, xenotime-(Y), REE-rich fluorapatite, pyrochlore group minerals Zr (700–21000 ppm, average 4700 ppm), Y (100–3900 ppm, average 550 ppm), Nb (40–400 ppm, average 180 ppm), REE (1000–1800 ppm, average 1300 ppm), Th (20–150 ppm), U (5–23 ppm) | **zircon**, **allanite-(Ce)**, **kainosite-(Y)**, **fergusonite-(Y)**, gadolinite-(Y), thorite, monazite-(Ce), REE-bearing titanite, britholite group minerals, *tengerite-(Y)*, *bastnäsite-(Y)* | **zircon**, **fergusonite-(Y)**, **britholite group minerals**, aeschynite-(Y), chevkinite-(Ce), yttrialite-(Y), thorite, monazite-(Ce), xenotime-(Y), *bastnäsite-(Ce)* ZrO$_2$ 6.5 wt %, Y$_2$O$_3$ 2.5 wt % REE (30000– 40000 ppm), Nb 14000 ppm, | **thorite**, **zircon**, **allanite-(Ce)**, **fergusonite-(Y)**, chevkinite-(Ce), monazite-(Ce), ferriallanite-(Ce), samarskite-(Y), aeschynite-(Y), Nb-bearing titanite, uraninite, fluorbritholite-(Ce), pyrochlore group minerals, gadolinite group minerals, *REE-carbonates* REE 85000 ppm, Y 42000 ppm, Nb 62000 ppm, Zr 67000 ppm, Th 17000 ppm, U 7700 ppm | **zircon**, **britholite group minerals**, pyrochlore group minerals Zr (10000–16000 ppm), Y 900 ppm, REE$_2$O$_3$ (0.1–0.3 wt %), Nb (up to 1200 ppm) | **britholite group minerals**, **REE-rich fluorapatite**, **hainite-(Y)**, **batievaite-(Y)**, **meliphanite**, leucophanite, zircon, pyrochlore group minerals, gadolinite group minerals, zirkelite, *cerianite-(Ce)*, *behoite* |

Note: Bold font is for major minerals; italic—for secondary minerals.

### 2.1. Occurrences Related to Keivy Alkali Granite

Mineralized granite bodies up to 1.5 × 4 km at the surface are confined to the apical parts of the alkali granite massifs (Figure 1). The rocks are silica-rich (up to 40–50 vol % of quartz) and enriched in Zr, Y, Nb, REE, Th, and U (Table 1).

The main REE- and actinide-bearing minerals are chevkinite-(Ce), bastnäsite-(Ce), allanite-(Ce), fergusonite-(Y), monazite-(Ce), britholite-(Y), thorite, xenotime-(Y), REE-rich fluorapatite, and pyrochlore group minerals. Rare-metal minerals have certain features of hydrothermal origin, namely a chain-like distribution of porous and metamict zircon, bud-shaped aggregates of zircon, alteration rims on anhedral "chevkinite" and "britholite", and the presence of REE-rich carbonate phases.

Dozens of pegmatite bodies from the Keivy province are confined to the inner and outer apical parts of alkali granite intrusions. They are a few tens meters long and several meters thick, sometimes forming "schlieren" and oval forms. Inner-chamber pegmatites are subdivided on the basis of quartz-microcline and quartz-feldspar-astrophyllite mineral types. The most abundant rare-element minerals are zircon, fergusonite-(Y), gadolinite-(Y), and thorite. Keivy pegmatites are of the gadolinite type with an Y, HREE, Zr, Ti, Nb > Ta, F signature indicating its NYF nature according to the Ćerny & Ercit classification [31]. The pegmatite studied here is from the White Tundra massif (Figure 1) and has a quartz-rich core rimmed by a pegmatoidal aegirine-arfvedsonite granite. The body is elongated (2 × 5 m). In addition to common rare-metal mineralization (zircon, fergusonite-(Y), gadolinite-(Y), thorite, monazite, allanite-(Ce), kainosite-(Y), astrophyllite, britholite-group minerals) the pegmatite is characterized by abundant galena and secondary REE mineralization (tengerite-(Y), bastnäsite-(Y)).

Quartzolites occur at the apical parts of alkali granite intrusions where they are hosted by both the granites (referred here as quartzolite-I) and country rocks (quartzolite-II). The Rova occurrence (Figure 1) is studied here. Most quartzolite bodies from Rova vary from 0.5 to 1.5 m across. They are inequigranular, taxitic rocks, ranging from medium-grained to pegmatitic. A distinguishing feature is the very low content of feldspar (<10 vol %). Quartz contents range from 50–90 vol %. The primary mafic rock-forming mineral is aegirine or arfvedsonite; occasionally they are present in equal amounts. Magnetite and ilmenite also frequently occur as rock-forming minerals but in lesser amounts. Sporadic large annite laths occur in some bodies. The significant amount of fluorite and other $F^-$ and $OH^-$ bearing minerals indicates the active involvement of fluids in the formation of the quartzolites. The whole rock may contain extremely high REE, Nb, Zr, and Y (Table 1).

The rare-metal mineralization in the Keivy quartzolites is volumetrically significant (up to 20–30 vol %) and variable in composition. Zircon is the typomorphic mineral of the ore assemblages. Other rare-metal minerals are irregularly distributed (from a few grams to tens of kilograms per ton of rock) and are represented by aeschynite-(Y), chevkinite-(Ce), fergusonite-(Y), britholite-group minerals, yttrialite-(Y), thorite, monazite-(Ce), xenotime-(Y), and bastnäsite-(Ce). Fluorbritholite-(Y), ytttrialite-(Y), fergusonite-(Y), and chevkinite-(Ce) are normally the main REE carriers and their overall content may reach 15–18 vol %.

The rare-metal minerals in quartzolite tend to form aggregates embedded in a quartz matrix. The minerals clearly are broadly coeval. Mutual contacts between the phases are sharp and occasionally one phase forms inclusions in another, such as zircon in monazite and fergusonite in chevkinite. The only minerals with a firmly established order of crystallization are the yttrialite—(Y) and later fluorbritholite—(Y).

The metasomatic rocks studied here, the El'ozero occurrence (Figure 1), are confined to a linear tectonic zone at the contact between the Keivy terrane and the Central Kola composite terrane. The zone strikes in a SE–NW direction for ca. 12 km and has an outcrop width varying from 200 to 1500 m. A few hundred alkali granite and aplite veins are confined to this zone and are concordant with the faults. The length of the veins is 50–500 m, with thicknesses of 3–50 m. The granites intruded a variety of rocks in the Keivy complex, namely, gabbro-anorthosites, gabbro-amphibolites, and gneisses. In some cases, the country rocks were metasomatically altered by hydrothermal fluids emanating from granite.

Metasomatic alteration of gabbro-anorthosite, associated with granite veins, ranges from unaltered rocks to more intensively altered types as follows; massive gabbro-anorthosite-plagioamphibolite-amphibole-biotite metasomatite-mineralized garnet-biotite metasomatite-mineralized biotite-albite metasomatite with quartz-(epidotic metasomatite). Rare-metal mineralization is confined to small linear and lensoid bodies, nodules and pods 0.5–2 m in size. The rocks are characterized by extremely high REE, Y, Nb, Zr, Th, and U (Table 1). Rare-metal mineralization is represented by thorite, chevkinite-(Ce), ferriallanite-(Ce) and allanite-(Ce), zircon, 'monazite', fergusonite-(Y), samarskite-(Y), aeschynite-(Y), Nb-bearing titanite and rutile, ilmenite, magnetite, cassiterite, REE-carbonates, uraninite, and pyrochlore and gadolinite-group minerals. It is assumed that rare-metal minerals crystallized under low-temperature conditions as most of them include zircon considered to be hydrothermal from the presence of porous, numerous mineral inclusions and the high content of Y, REE, Hf. The sequences of crystallization and the compositions of the rare-metal minerals indicate that the metasomatites formed under the influence of alkaline, F-bearing fluids with high activities of $CO_2$, Si, Ca, and Al at some stages [32].

## 2.2. Occurrences Related to Sakharjok Nepheline Syenite

A complex Zr-REE deposit is confined to the Sakharjok alkaline rock massif (Figure 1). The massif is a fissure-type intrusion 7 km long and 4–5 $km^2$ at outcrop. It intrudes the Late Archean West Keivy alkali granite and gneiss-diorites of the TTG basement. The massif is composed of alkali feldspar syenite and nepheline syenite of 2682 ± 10 Ma and 2613 ± 35 Ma age, respectively [30]. These ages coincide within error limits with the age of the adjacent Keivy alkali granite (2654 ± 5 Ma [30]), which suggests their origin during the same magmatic event. The western and southwestern parts of the massif are made up of lepidomelane-ferrohastingsite syenites, while the eastern part consists of trachytoid lepidomelane-aegirine-augite nepheline syenites. The nepheline syenites contain large (up to 80 × 200 m) exposures of metasomatised basic rocks and lenticular bodies of porphyritic nepheline syenite. The latter was formed by recrystallization of trachytoid nepheline syenites with the formation of large poikilitic ferrohastingsite phenocrysts after aegirine and lepidomelane. At the same time, there are no significant chemical differences between these rock types. The characteristic feature of the nepheline syenites is the presence of small (up to 1–2 m) and morphologically diverse (schlieren, veinlets, and irregularly shaped segregations) pegmatoidal bodies. In mineral composition, they are generally similar to the nepheline syenite, except for an enrichment in accessory minerals (zircon, britholite and pyrochlore group minerals, fluorite, galena, and others).

A mineralized syenite is located in the northern part of the exposed nepheline syenites. An ore block ca 2 $km^2$ in area consists of several linear bodies from 200 to 1350 m long and 3–30 m thick. Based on drilling, the bodies dip steeply at 70–80° northeastward. Their orientation in general is conformable with the primary magmatic banding and trachytoid texture of the nepheline syenite, suggesting that the ore bodies were formed at the magmatic stage during intrachamber differentiation [27]. From petrography, the ores are similar to the barren nepheline syenite, but differ in the elevated contents of zircon (0.5–1.2 vol %, locally up to 2.5%) and britholite (0.2–1.0 vol %).

It was previously shown [27] that nepheline syenite of the Sakharjok massif is enriched in high field strength cations and volatiles: Zr 1000–5000 ppm, Y 100–500 ppm, $REE_2O_3$ 0.1–0.3 wt %, Nb 150–1200 ppm, and F 0.1–1.2 wt %. Significant enrichment of the nepheline syenite in REE with respect to chondrites (in general 50–1000 times LREE and 50–200 times HREE), a steep slope of REE distribution ((La/Yb)n varies from 3.5 to 12, average 5.9) in combination with the negative Eu anomaly (average Eu/Eu* 0.29) indicate a significant role for fractional crystallization of a parental alkali basaltic magma during formation of the rocks of the Sakharjok massif. Albitization and hydrothermal mineral formation at the late- and post-magmatic stages of the massif formation have been established by petrographic study of rocks and the zircon mineralogy [27,33,34].

The nepheline syenite pegmatite studied here occurs within the contact zone between nepheline syenite and basic rock, outcrops up to 30 $m^2$ in area, and consists of nepheline, albite, pyroxenes

(mainly aegirine-augite), amphiboles (mainly ferrohastingsite), biotite, and analcime. Other minor and accessory minerals observed in the pegmatite are: apatite supergroup minerals, batievaite-(Y), behoite, calcite, cerianite-(Ce), fluorite, gadolinite-subgroup minerals, hainite-(Y), ilmenite, leucophanite, meliphanite, mimetite, molybdenite, nickeline, pyrochlore group minerals, rutile, smectite, titanite, thomsonite-Ca, zircon, and zirkelite [33,35].

A metasomatic rock occurs at the contact between nepheline syenite and basic rock. It is an elongated body ca. 1–1.5 m thick, composed mainly of coarse-grained biotite (up to 90 vol %), secondary nepheline and plagioclase and accessory fluorite and apatite supergroup minerals.

## 3. Analytical Methods

The chemical compositions of BGM were determined by electron microprobe analysis in Geological Institute, Kola Science Centre, Apatity, and in the Inter-Institute Analytical Complex at the Institute of Geochemistry, Mineralogy, and Petrology, University of Warsaw. Cameca MS-46 (Cameca, Paris, France) was used in Apatity (WDS (wavelength-dispersive spectrometry) mode, 22 kV, 30–40 nA, counting times were five measurements on 10 s each on peak and 10 s on each of two background positions). Fluorine content was determined using LEO-1450 SEM (scanning electron microscope, Carl Zeiss AG, Oberkochen, Germany) equipped with an XFlash-5010 Bruker Nano GmbH EDS (Bruker Nano GmbH, Berlin, Germany). The electron microscope operated at acceleration voltage 20 kV, current intensity 0.5 nA, accumulation time 200 s, procedure of standard-free analysis by the P/B–ZAF method of the QUANTAX system.

Mineral compositions obtained at University of Warsaw were determined using a Cameca SX-100 microprobe (Cameca, Paris, France) equipped with four WDS detectors. The accelerating voltage was 15 kV and the probe current was 20 nA. Counting times were 20 s on peak and 10 s on each of two background positions. The 'PAP' and 'X-PHI' $\varphi(\rho z)$ models [36,37] were used for corrections.

The standards, crystals, and X-ray lines used and detection limits for both Laboratories are given in Supplementary Table S1. Representative electron microprobe data for the BGM are given in Table 2; the full data set for BGM (90 EPMA analyses) is presented in Supplementary Table S2.

**Table 2.** Representative chemical compositions and mineral formulae of britholite group minerals from REE-rich lithologies related to the Keivy alkali granite—nepheline syenite complex.

| Sample | 172 | 9 | 2 | 9/1 | 162/60 | 302 | 300 | 529 | 532 | 633 |
|---|---|---|---|---|---|---|---|---|---|---|
| wt % | | | | | | | | | | |
| $P_2O_5$ | 4.88 | 5.15 | 0.00 | 2.23 | b.d. | 3.38 | 1.60 | 11.86 | 10.57 | 4.84 |
| $SiO_2$ | 21.92 | 19.75 | 23.17 | 22.74 | 21.09 | 21.52 | 20.91 | 16.16 | 18.14 | 19.80 |
| $ThO_2$ | 0.78 | 0.75 | b.d. | b.d. | b.d. | 2.85 | 0.85 | 0.27 | b.d. | 2.44 |
| $UO_2$ | b.d. | 0.28 | b.d. | 0.12 | b.d. | b.d. | b.d. | b.d. | b.d. | b.d. |
| $SO_2$ | n.a. | n.a. | n.a. | b.d. | n.a. | n.a. | n.a. | n.a. | n.a. | 0.46 |
| $TiO_2$ | n.a. | n.a. | n.a. | b.d. | 0.76 | n.a. | n.a. | n.a. | n.a. | n.a. |
| $ZrO_2$ | n.a. | n.a. | n.a. | n.a. | n.a. | n.a. | n.a. | n.a. | n.a. | 0.88 |
| $Al_2O_3$ | b.d. | b.d. | b.d. | b.d. | b.d. | 0.28 | 0.05 | 0.07 | 0.15 | 0.17 |
| $Y_2O_3$ | 30.64 | 18.11 | 41.11 | 28.11 | 0.48 | 10.53 | 10.34 | 10.36 | 12.40 | 10.61 |
| $La_2O_3$ | 1.61 | 3.32 | 0.34 | 2.57 | 18.82 | 8.52 | 13.88 | 8.73 | 9.10 | 7.87 |
| $Ce_2O_3$ | 4.25 | 9.94 | 1.46 | 7.01 | 31.26 | 15.66 | 18.91 | 11.93 | 13.37 | 14.85 |
| $Pr_2O_3$ | 0.45 | 1.83 | 0.13 | 0.58 | 2.40 | 1.52 | 1.25 | 0.78 | 0.98 | 1.43 |
| $Nd_2O_3$ | 1.80 | 7.36 | 0.85 | 3.68 | 9.08 | 6.08 | 4.39 | 2.80 | 3.04 | 5.09 |
| $Sm_2O_3$ | 0.51 | 2.23 | 1.00 | 1.41 | 0.75 | 1.24 | 0.89 | 0.59 | 0.67 | 1.12 |
| $Gd_2O_3$ | 1.07 | 3.30 | 1.77 | 3.74 | 3.49 | 1.33 | 1.25 | 0.70 | 0.62 | 1.37 |
| $Tb_2O_3$ | 0.28 | 0.32 | 0.55 | b.d. | b.d. | b.d. | b.d. | b.d. | b.d. | b.d. |
| $Dy_2O_3$ | 3.12 | 2.28 | 5.78 | 3.37 | 0.23 | 1.68 | 1.70 | 0.74 | 1.00 | 1.56 |
| $Ho_2O_3$ | 0.70 | 0.65 | 1.39 | 0.11 | b.d. | 0.20 | 0.28 | b.d. | 0.48 | b.d. |
| $Er_2O_3$ | 3.26 | b.d. | 5.33 | b.d. | b.d. | 1.23 | 1.48 | 0.80 | 0.91 | 1.07 |
| $Tm_2O_3$ | 0.57 | b.d. | 0.78 | b.d. | b.d. | 0.11 | 0.14 | 0.19 | 0.22 | 0.24 |
| $Yb_2O_3$ | 2.35 | 1,17 | 3.79 | 2.35 | b.d. | 1.33 | 1.57 | 1.22 | 1.21 | 1.24 |

**Table 2.** *Cont.*

| Sample | 172 | 9 | 2 | 9/1 | 162/60 | 302 | 300 | 529 | 532 | 633 |
|---|---|---|---|---|---|---|---|---|---|---|
| $Lu_2O_3$ | 0.25 | b.d. | 0.38 | 0.56 | b.d. | 0.23 | 0.14 | b.d. | 0.08 | 0.25 |
| FeO | 0.71 | 0.36 | 0.88 | 0.50 | 3.36 | 0.05 | 0.17 | b.d. | 0.05 | 0.07 |
| MnO | 1.82 | 1.15 | 0.58 | 0.33 | 0.10 | 0.06 | 0.67 | 0.05 | b.d. | 0.17 |
| CaO | 16.48 | 15.72 | 4.21 | 14.61 | 5.36 | 17.35 | 12.45 | 26.08 | 24.63 | 18.86 |
| SrO | 0.25 | 0.04 | b.d. | b.d. | n.a. | n.a. | n.a. | b.d. | b.d. | b.d. |
| BaO | n.a. | 0.00 | n.a. | 0.11 | n.a. | n.a. | n.a. | n.a. | n.a. | n.a. |
| PbO | n.a. | n.a. | n.a. | 0.37 | n.a. | 0.30 | b.d. | b.d. | b.d. | b.d. |
| $Na_2O$ | 0.19 | b.d. | 1.70 | 0.16 | b.d. | b.d. | 0.18 | 0.09 | 0.08 | 0.18 |
| $K_2O$ | b.d. | b.d. | b.d. | b.d. | b.d. | b.d. | 0.04 | b.d. | b.d. | b.d. |
| F | 2.08 | 1.29 | 1.16 | n.a | 1.18 | 1.36 | 1.65 | 1.72 | 1.26 | 1.28 |
| Cl | b.d. | b.d. | b.d. | 0.04 | b.d. | b.d. | b.d. | b.d. | b.d. | 0.05 |
| Sum | 99.98 | 95.00 | 96.37 | 98.58 | 98.48 | 96.80 | 94.77 | 95.14 | 98.98 | 95.90 |
| O = -F | 0.88 | 0.54 | 0.49 | 1.60 | 0.50 | 0.57 | 0.69 | 0.72 | 0.53 | 0.54 |
| O = -Cl | 0.00 | 0.00 | 0.00 | 0.01 | 0.00 | 0.00 | 0.00 | 0.00 | 0.00 | 0.01 |
| Total | 99.10 | 94.46 | 95.88 | 96.98 | 97.98 | 96.22 | 94.08 | 94.41 | 98.45 | 95.35 |
| Formulae on the basis of 8 total (M + T) cations | | | | | | | | | | |
| P | 0.47 | 0.55 | 0.00 | 0.23 | 0.00 | 0.36 | 0.18 | 1.14 | 1.00 | 0.51 |
| Si | 2.52 | 2.49 | 3.00 | 2.79 | 3.09 | 2.70 | 2.84 | 1.84 | 2.02 | 2.44 |
| Sum T | 2.99 | 3.04 | 3.00 | 3.02 | 3.09 | 3.06 | 3.02 | 2.98 | 3.02 | 3.00 |
| Th | 0.02 | 0.02 | 0.00 | 0.00 | 0.00 | 0.08 | 0.03 | 0.01 | 0.00 | 0.07 |
| U | 0.00 | 0.01 | 0.00 | 0.00 | 0.00 | 0.00 | 0.00 | 0.00 | 0.00 | 0.00 |
| S | 0.00 | 0.00 | 0.00 | 0.00 | 0.00 | 0.00 | 0.00 | 0.00 | 0.00 | 0.05 |
| Ti | 0.00 | 0.00 | 0.00 | 0.00 | 0.08 | 0.00 | 0.00 | 0.00 | 0.00 | 0.00 |
| Zr | 0.00 | 0.00 | 0.00 | 0.00 | 0.00 | 0.00 | 0.00 | 0.00 | 0.00 | 0.05 |
| Al | 0.00 | 0.00 | 0.00 | 0.00 | 0.00 | 0.04 | 0.01 | 0.01 | 0.02 | 0.03 |
| Y | 1.87 | 1.21 | 2.84 | 1.84 | 0.04 | 0.70 | 0.75 | 0.63 | 0.74 | 0.70 |
| La | 0.07 | 0.15 | 0.02 | 0.12 | 1.02 | 0.39 | 0.70 | 0.37 | 0.37 | 0.36 |
| $Ce^{3+}$ | 0.18 | 0.46 | 0.07 | 0.31 | 1.68 | 0.72 | 0.94 | 0.50 | 0.55 | 0.67 |
| Pr | 0.02 | 0.08 | 0.01 | 0.03 | 0.13 | 0.07 | 0.06 | 0.03 | 0.04 | 0.06 |
| Nd | 0.07 | 0.33 | 0.04 | 0.16 | 0.48 | 0.27 | 0.21 | 0.11 | 0.12 | 0.22 |
| Sm | 0.02 | 0.10 | 0.04 | 0.06 | 0.04 | 0.05 | 0.04 | 0.02 | 0.03 | 0.05 |
| Gd | 0.04 | 0.14 | 0.08 | 0.15 | 0.17 | 0.06 | 0.06 | 0.03 | 0.02 | 0.06 |
| Tb | 0.01 | 0.01 | 0.02 | 0.00 | 0.00 | 0.00 | 0.00 | 0.00 | 0.00 | 0.00 |
| Dy | 0.12 | 0.09 | 0.24 | 0.13 | 0.01 | 0.07 | 0.07 | 0.03 | 0.04 | 0.06 |
| Ho | 0.03 | 0.03 | 0.06 | 0.00 | 0.00 | 0.01 | 0.01 | 0.00 | 0.02 | 0.00 |
| Er | 0.12 | 0.00 | 0.22 | 0.00 | 0.00 | 0.05 | 0.06 | 0.03 | 0.03 | 0.04 |
| Tm | 0.02 | 0.00 | 0.03 | 0.00 | 0.00 | 0.00 | 0.01 | 0.01 | 0.01 | 0.01 |
| Yb | 0.08 | 0.04 | 0.15 | 0.09 | 0.00 | 0.05 | 0.07 | 0.04 | 0.04 | 0.05 |
| Lu | 0.01 | 0.00 | 0.01 | 0.02 | 0.00 | 0.01 | 0.01 | 0.00 | 0.00 | 0.01 |
| $Fe^{2+}$ | 0.07 | 0.04 | 0.09 | 0.05 | 0.41 | 0.00 | 0.02 | 0.00 | 0.01 | 0.01 |
| $Mn^{2+}$ | 0.18 | 0.12 | 0.06 | 0.03 | 0.01 | 0.01 | 0.08 | 0.00 | 0.00 | 0.02 |
| Ca | 2.03 | 2.12 | 0.58 | 1.92 | 0.84 | 2.33 | 1.81 | 3.18 | 2.94 | 2.49 |
| Sr | 0.02 | 0.00 | 0.00 | 0.00 | 0.00 | 0.00 | 0.00 | 0.00 | 0.00 | 0.00 |
| Ba | 0.00 | 0.00 | 0.00 | 0.01 | 0.00 | 0.00 | 0.00 | 0.00 | 0.00 | 0.00 |
| Pb | 0.00 | 0.00 | 0.00 | 0.01 | 0.00 | 0.01 | 0.00 | 0.00 | 0.00 | 0.00 |
| Na | 0.04 | 0.00 | 0.43 | 0.04 | 0.00 | 0.00 | 0.05 | 0.02 | 0.02 | 0.04 |
| K | 0.00 | 0.00 | 0.00 | 0.00 | 0.00 | 0.00 | 0.01 | 0.00 | 0.00 | 0.00 |
| Sum M | 5.01 | 4.96 | 5.00 | 4.98 | 4.90 | 4.94 | 4.98 | 5.02 | 4.98 | 5.00 |
| F | 0.76 | 0.51 | 0.47 | | 0.55 | 0.54 | 0.71 | 0.62 | 0.44 | 0.50 |
| Cl | 0.00 | 0.00 | 0.00 | | 0.00 | 0.00 | 0.00 | 0.00 | 0.00 | 0.01 |
| OH | 0.13 | 0.16 | 0.12 | | | 0.31 | 0.07 | 0.38 | 0.52 | 0.49 |
| O | 0.11 | 0.33 | 0.41 | | 0.45 | 0.15 | 0.22 | | 0.04 | |
| Sum X | 1.00 | 1.00 | 1.00 | | 1.00 | 1.00 | 1.00 | 1.00 | 1.00 | 1.00 |

172—fluorbritholite-(Y), quartzolite-I; 9—fluorbritholite-(Y), quartzolite-II; 2—britholite-(Y), granitic pegmatite; 9/1—britholite-(Y), granitic pegmatite; 162/60—fluorbritholite-(Ce), quartz-epidote metasomatite; 302—fluorbritholite-(Ce), porphyritic nepheline syenite; 300—fluorbritholite-(Ce), trachytoid nepheline syenite; 529—fluorcalciobritholite, nepheline syenite pegmatite; 532—'calciobritholite', nepheline syenite pegmatite; 633—britholite-(Y), micaceous metasomatite. b.d.—below detection limit; n.a.—not analyzed.

## 4. Results

### 4.1. Morphology, Internal Textures, Associated Minerals, and Alteration

BGM from quartzolite bodies are abundant, reaching 5–10 vol %. Three morphological types are distinguished in quartzolite of the endocontact zone: (1) subhedral (most abundant) grains up to 1–2 cm (Figure 2a,b); (2) anhedral grains in intergrowths with yttrialite-(Y) (Figure 2c); (3) poikilitic crystals and skeletal aggregates. BGM from quartzolite-I have a thin alteration rim which is composed of an X-ray amorphous substance with lower Ca and Y contents and with inclusions of monazite-(Ce) and bastnäsite-(Ce).

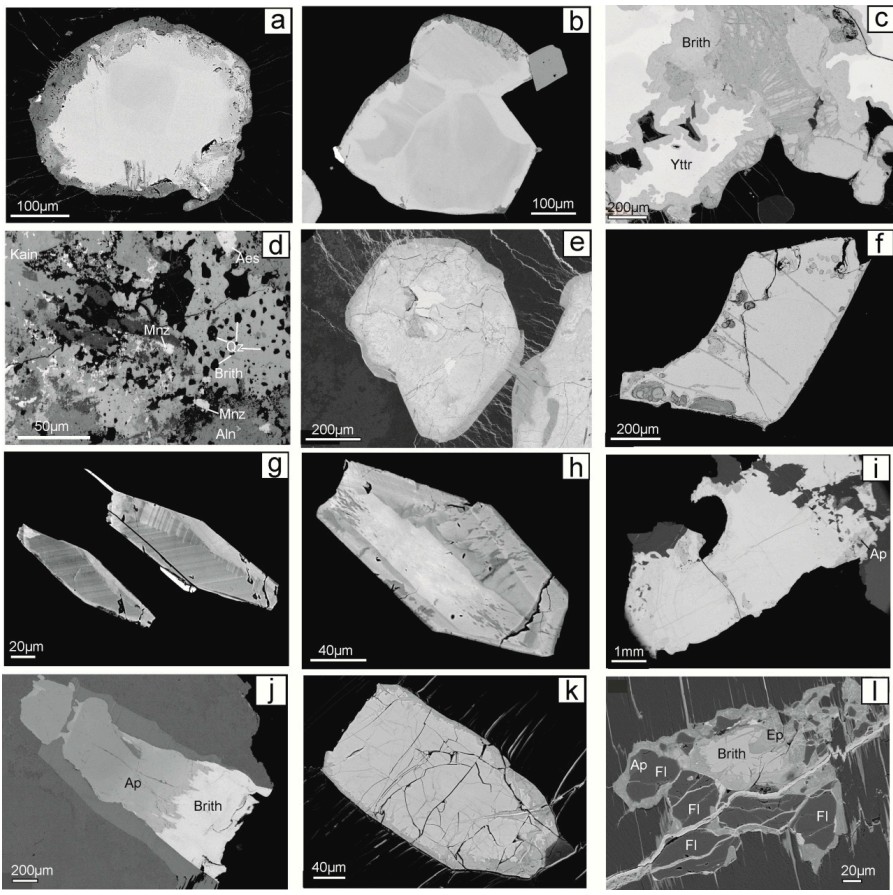

**Figure 2.** BSE images showing morphologies and internal textures of BGM and associated minerals from REE-rich lithologies related to the Keivy alkali granite-nepheline syenite complex: (**a**,**b**)—subhedral zoned crystals of fluorbritholite-(Y) with alteration rims from quartzolite-I; (**c**) fluorbritholite-(Y) overgrowths after yttrialite-(Y) in quartzolite; (**d**)—euhedral crystal of britholite-(Y) with inclusions of quartz, allanite-(Ce) and monazite-(Ce) from alkali granite pegmatite; (**e**)—patchily zoned subhedral crystal of britholite-(Ce) with alteration rims from porphyritic nepheline syenite; (**f**)—subhedral crystal of britholite-(Ce) with alteration rims from trachytoid nepheline syenite; (**g**)—euhedral rhythmic zoned crystal of fluorcalciobritholite from nepheline syenite pegmatite; (**h**)—euhedral zoned crystal of fluorcalciobritholite from nepheline syenite pegmatite; (**i**)—fluorbritholite-(Ce) with inclusions of fluorapatite and K-feldspar, nepheline syenite pegmatite; (**j**)—intergrowths of fluorcalciobritholite and fluorapatite from nepheline syenite pegmatite; (**k**)—euhedral crystal of britholite-(Y) with alteration rims from micaceous metasomatite in nepheline syenite; (**l**)—intergrowth of anhedral britholite-(Y), fluorite and fluorapatite in micaceous metasomatite. Brith—britholite group mineral; Yttr—yttrialite-(Y); Ap—fluorapatite; Ep—epidote; Fl—fluorite; Aln—allanite-(Ce); Mnz—monazite-(Ce); Kain—kainosite-(Y); Aes—aeschynite-(Y); Qz—quartz.

The BGM from the alkali granite pegmatite occurs as rare subhedral and anhedral, 1–2 mm poikilitic grains in association with allanite-(Ce) and kainosite-(Y) (Figure 2d). It may contain numerous inclusions of quartz, allanite-(Ce), monazite-(Ce), aeschynite-(Y), and titanite.

The BGM from a quartz-epidote (apobasic) metasomatite form up to 10 μm anhedral grains and veins of 2–5 μm thickness in rutile-magnetite symplectite which along with ilmenite replace ferriallanite-(Ce) [32].

The BGM from the mineralized granites occur within clusters of rare-metal minerals, among which are zircon, thorite, fergusonite-(Y), and REE-rich fluorapatite. Britholite-(Y) forms rims up to 100 μm around anhedral fluorapatite. The mineral has undergone alteration with the formation of thin rims, composed of a metamict phase containing Ca, REE, Pb, and actinides [38].

The BGM from the mineralized nepheline syenite are represented by large (up to 20 mm) aggregates of grains in recrystallized amphibole (porphyritic) syenite (Figure 2e) and by individual (up to 1 mm) subhedral and anhedral crystals in lepidomelane-aegirine syenite (Figure 2f). The mineral often includes 'porous' zircon, fluorite, and albite, which points to a late- and post-magmatic origin. BGM from nepheline syenites are altered, with the formation of lower Z (in BSE) patches and rims and the release of F, Ce, and La. A possible reaction for rim formation is britholite + fluid → apatite + epidote + REE carbonate.

The BGM from the nepheline syenite pegmatite tend to form single crystals, rare aggregates and intergrowths with fluorapatite embedded in a nepheline-feldspar-mica-pyroxene-zeolite matrix. Fluorcalciobritholite is the most abundant BGM species. Usually, it forms single euhedral, rarely subhedral, crystals of dipyramidal-prismatic habit (Figure 2g,h). Aggregates of several crystals also occur. The mineral is characterized by heterogeneous internal textures of three types: patchy, rhythmic, and combined patchy-rhythmic, indicating changes of the crystallization environment. The most abundant mineral inclusions in fluorcalciobritholite are fluorapatite, potassium feldspar, and more rarely pyroxene and monazite-(Ce). Anhedral grains of fluorcalciobritholite are rare. 'Calciobritholite' forms either irregular patches within the euhedral crystals of fluorcalciobritholite or rims on subhedral individuals of fluorcalciobritholite. Fluorbritholite-(Y) occurs in polyphase inclusions with fluorapatite and silicate minerals in euhedral crystals of fluorcalciobritholite. The samples with fluorbritholite-(Y) contain abundant disseminated fluorite. Fluorbritholite-(Ce) forms large (up to 1 cm) anhedral grains (Figure 2i). The mineral has a heterogeneous internal structure, numerous inclusions of apatite and minor potassium feldspar. The outer rim of fluorbritholite-(Ce) is often intensively altered.

The common occurrence of fluorcalciobritholite and fluorapatite intergrowths in nepheline syenite pegmatite points to a genetic link. Macroscopically, the intergrowths look like elongated subhedral single crystals and minerals are distinguished only by SEM study. The boundaries between the two phases are always sharp and of 'sawtooth' shape (Figure 2j). Fluorcalciobritholite from intergrowths has a patchy internal texture, and fluorapatite has a coarse zoning expressed in successive increases of the britholite component (higher BSE intensity) towards the rim.

From the textural relationships, it can be firmly concluded that the fluorbritholite-(Y) is apparently the earliest BGM in the nepheline syenite pegmatite. Fluorcalciobritholite is the typomorphic mineral in the pegmatite assemblage. It can be suggested that fluorcalciobritholite with rhythmic texture crystallized before patchily zoned fluorcalciobritholite. Calciobritholite formed at different stages but, in greater part, is associated with patchy fluorcalciobritholite and after fluorcalciobritholite with rhythmic texture. Fluorbritholite-(Ce) apparently formed only during the latest stages.

Britholite-(Y) from micaceous metasomatite forms two morphological types: (1) euhedral crystals (Figure 2k); (2) anhedral grains intergrown with fluorite, fluorapatite, and REE-bearing phases (Figure 2l). Minerals of both types have alteration rims of Ca-Al-Fe-silicate (possibly epidote).

### 4.2. BGM Composition and Mineral Species

The Keivy BGM are represented by most of the end-members according to the IMA-approved classification of Pasero et al. [25]: fluorbritholite-(Y), britholite-(Y), fluorbritholite-(Ce), britholite-(Ce), and fluorcalciobritholite. In addition, several samples of the hypothetical "calciobritholite" were identified. The identification of all analyzed grains is provided in Supplementary Table S2; representative BGM compositions and mineral formulae are given in Table 2.

Y-dominant species are characteristic of lithologies related to alkali granite (quartzolites, pegmatite) and in nepheline syenite pegmatite (Figure 3). BGM from the trachytoid nepheline syenite and single grains from the nepheline syenite pegmatite belong to Ce-dominant species. The maximum Ce component is observed in BGM from a quartz-epidote (apobasic) metasomatite. BGM from porphyritic nepheline syenite are both Y- and Ce-species, whereas the mineral from the micaceous metasomatite has equal proportions of Ce and Y.

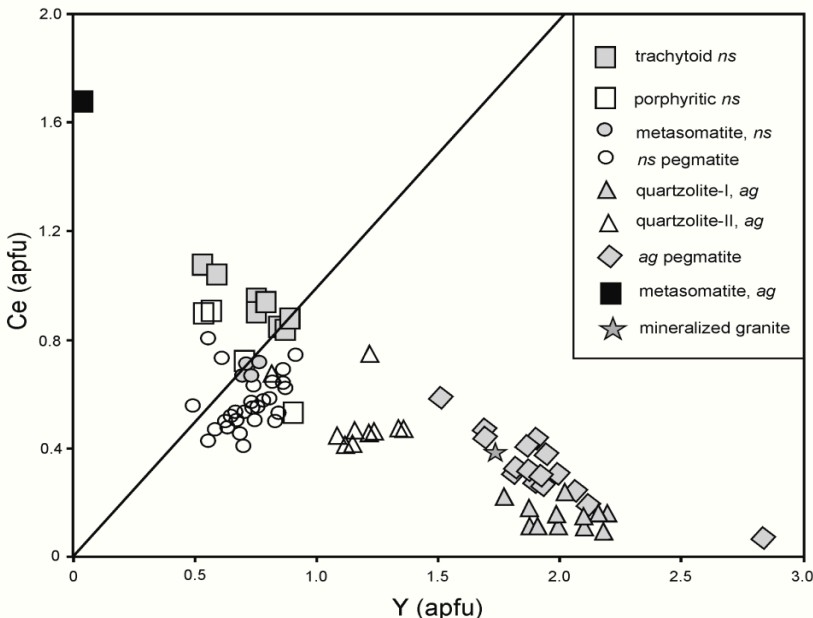

**Figure 3.** Y vs. Ce plot for BGM showing the distribution of Y- and Ce-dominant species in the REE-rich lithologies of the Keivy alkali granite—nepheline syenite complex. *ag*—alkali granite; *ns*—nepheline syenite. Figure shows that the alkaline granite BGM and the nepheline syenite BGM have distinctly different compositions and discrete populations.

In fluorbritholite-(Y) from quartzolite-I the M site has Ca (1.56–2.10 apfu) and REE (2.62–3.05 apfu), the T site is dominated by Si (2.5–3.0 apfu), with P (0–0.53 apfu). Fluorbritholite-(Y) and britholite-Y) from quartzolite-II have these values as 1.81–2.42, 2.39–3.02, 2.3–2.9, 0.12–0.80 apfu, respectively. Most britholite-(Y) from the granitic pegmatite has values of Ca around 1.8 apfu, REE—3.0 apfu, Si—2.8–2.9 apfu, P—0.1–0.2 apfu, with one exception of extremely 'britholitic' composition (Ca = 0.58 apfu, REE = 3.82 apfu, Si = 3.0 apfu, P = 0 apfu). The same extremely 'britholitic' composition is found in fluorbritholite-(Ce) from the quartz-epidote (apobasic) metasomatite (Ca = 0.84 apfu, REE = 3.55 apfu, Si = 3.1 apfu, P = 0 apfu).

Britholite-(Ce) and fluorbritholite-(Ce) from the porphyritic / trachytoid mineralized nepheline syenite contain Ca (2.13–2.33/1.77–1.88 apfu), REE (2.46–2.51/2.91–3.02 apfu), Si (2.70–2.86/2.78–2.96 apfu) and P (0.22–0.36/0.12–0.20 apfu). Britholite-(Y) from micaceous metasomatite is different from the same minerals in quartzolite and granitic pegmatite in the elevated Ca (2.26–2.49 apfu) and P (0.31–0.51 apfu), and lower REE (2.29–2.49 apfu) and Si (2.44–2.69 apfu).

BGM from the nepheline syenite pegmatite are characterized by less 'britholitic' compositions. In fluorcalciobritholite and 'calciobritholite' from pegmatite, the M site is dominated by Ca (2.7–3.5 apfu), with REE (1.5–2.3 apfu), the T site has Si (1.5–2.3 apfu), with P (0.7–1.3 apfu).

The fluorine contents in Keivy BGM are as different in the various occurrences as in the same lithology, probably indicating a change of F activity in the crystallization media. BGM from quartzolite-I are fluorbritholite-(Y) with F in the X site (0.60–0.87 apfu). Both fluorbritholite-(Y) and britholite-(Y) occur in quartzolite-II with F ranging from 0.46–0.54 apfu. Similar F values (close to the 0.5 boundary) are observed for britholite-(Y) from the alkali granite pegmatite (0.47 apfu) and fluorbritholite-(Ce) from the quartz-epidote metasomatite (0.55 apfu). The changing F contents are also determined for BGM from nepheline syenites (0.27–0.68 apfu for porphyritic rocks, and 0.45–0.83 apfu for trachytoid types). The X site in the nepheline syenite pegmatite BGM is normally dominated by F in fluorcalciobritholite (0.5–0.7 apfu). As the F values for pegmatitic 'calciobritholite' are 0.38–0.48 apfu (i.e., close to the 0.5 boundary) coupled with the tight spatial association of the latter with fluorcalciobritholite we can refer them to the same genetic group. BGM from the micaceous metasomatite in nepheline syenite are consistently OH-dominant (0.37–0.5 F apfu) indicating a low activity of F, which was earlier consumed in the abundant fluorite.

The chlorine content is below detection limit in most Keivy BGM and only minerals from the micaceous metasomatite in nepheline syenite contain Cl, at levels of 0.01–0.02 apfu.

Minor elements—Fe, Mn, Th, Na, Sr, and Al—in BGM usually range from 0 to 0.04 apfu with the exception of Na (up to 0.06 apfu in quartzolitic BGM; 0.09 apfu in nepheline syenite BGM), Mn (0.1–0.18 apfu in alkali granite lithologies; 0.06–0.09 apfu in BGM from trachytoid BGM), Fe (0.06–0.09 apfu in quartzolite-I BGM) and Th (0.06–0.1 apfu in BGM from the porphyritic nepheline syenite and micaceous metasomatite).

Rare earth elements and yttrium in Keivy BGM mostly show values of $10^5$–$10^6$ times chondritic (Figure 4). The distribution of REE is as different between mineral species as between host lithologies.

The chondrite-normalized REE patterns for BGM from quartzolite are different (Figure 4A): britholites from quartzolite-I show a sinusoidal REE pattern with minima at Nd and Sm and maxima at Dy, Y, Er; britholites from quartzolite-II have a more or less straightforward pattern. Both groups of BGM have an unfractionated distribution of LREE ($La/Gd_n$ = 0.6–2.3, average 1.04). The distribution of HREE is different: unfractionated for quartzolite-I BGM and fractionated for quartzolite-II BGM (average $Gd/Yb_n$ are 0.3 and 2.3, respectively). As a whole, REE are unfractionated for britholites from quartzolite-I (average $La/Yb_n$ = 0.3) and fractionated for minerals from quartzolite-II (average $La/Yb_n$ = 2.4). Britholite-(Y) from the mineralized granite has an unfractionated pattern for the whole REE spectrum with $La/Gd_n$ = 0.46 and $Ce/Y_n$ = 1.0 (Figure 4C).

The chondrite-normalized REE patterns for the BGM from the granitic pegmatite (Figure 4B) are similar to those for quartzolite-I ($La/Gd_n$ = 0.2–1.6, average 0.73; $Gd/Yb_n$ = 0.4–1.6, average 0.98; average $La/Yb_n$ = 0.7). Of interest is one sample with an extremely unfractionated pattern ($La/Gd_n$ = 0.16; $Gd/Yb_n$ = 0.37; $La/Yb_n$ = 0.06). In turn, fluorbritholite-(Ce) from the quartz-epidote metasomatite shows an extremely fractionated pattern with $La/Gd_n$ = 4.5 and $Ce/Y_n$ = 180 (Figure 4C).

REE distributions in BGM from porphyritic and trachytoid nepheline syenites have more or less similar patterns (strongly fractionated for LREE and unfractionated for HREE (Figure 4D)). Nevertheless, the parameters are slightly different (BGM from porphyritic nepheline syenite have $La/Gd_n$ = 4.5–5.3 (average 4.9), $Gd/Yb_n$ = 0.7–1.4 (average 1.01) and average $La/Yb_n$ = 4.9; and for BGM from trachitoid syenite $La/Gd_n$ = 5.7–9.6 (average 7.6), $Gd/Yb_n$ = 0.6–1.9 (average 0.9) and average $La/Yb_n$ = 6.4). Britholite-(Ce) and fluorbritholite-(Ce) from trachytoid syenite have deep Y minima. The REE patterns for BGM from the micaceous metasomatite (Figure 4F) are similar to those from the porphyritic nepheline syenite ($La/Gd_n$ = 4.7–6.0 (average 5.5), $Gd/Yb_n$ = 0.67–0.88 (average 0.78) and average $La/Yb_n$ = 4.3).

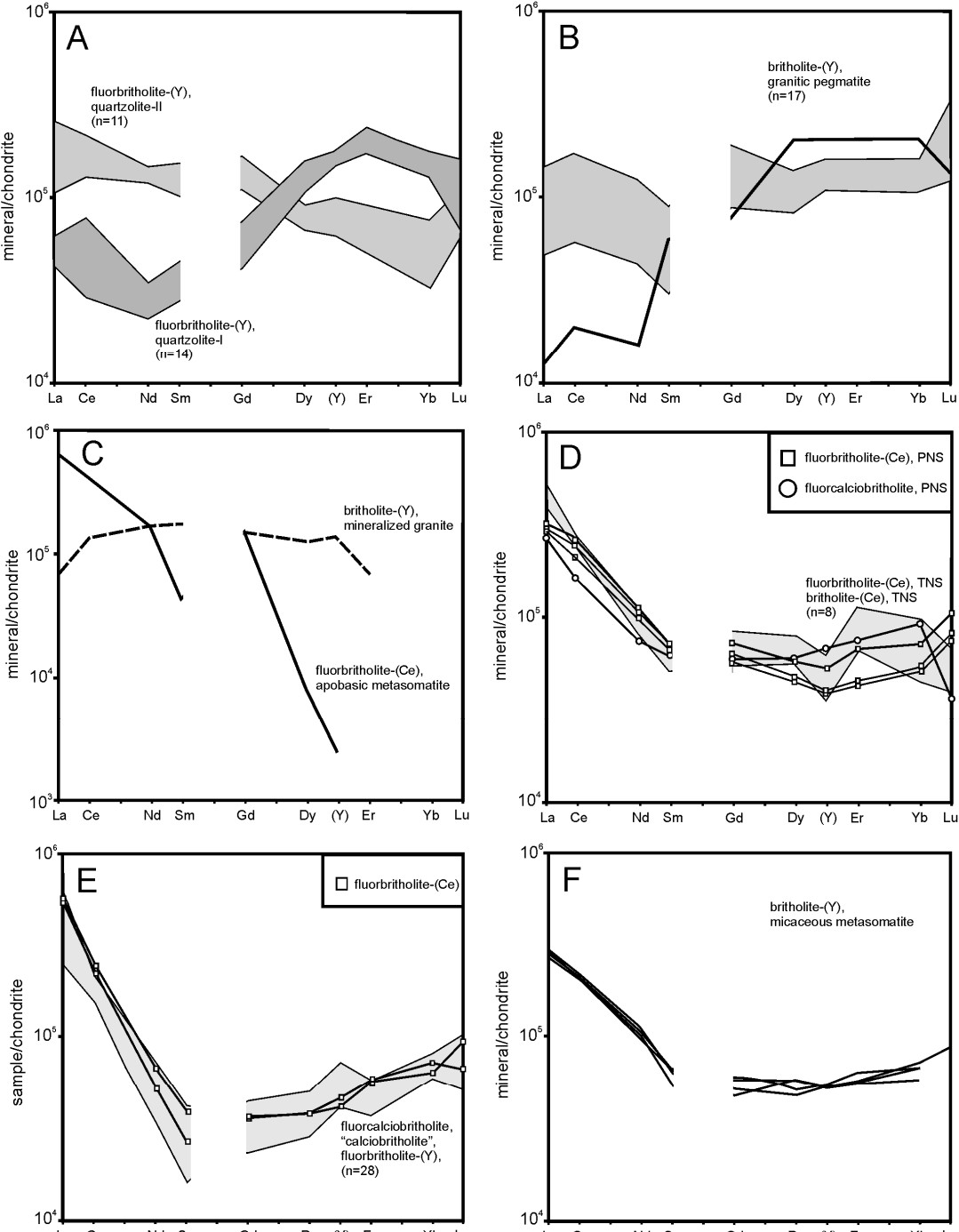

**Figure 4.** Chondrite-normalized REE patterns for BGM from REE-rich lithologies related to the Keivy alkali granite-nepheline syenite complex, with Y proxying for Ho: (**A**)—quartzolites; (**B**)—alkali granite pegmatite (solid line is for britholite-(Y) with extremely unfractionated REE distribution); (**C**)—mineralized granite and quartz-epidote (apobasic) metasomatite; (**D**)—porphyritic (PNS) and trachytoid (TNS) nepheline syenites; (**E**)—nepheline syenite pegmatite; (**F**)—micaceous metasomatite in nepheline syenite. Normalizing factors from [39]. Figure 4 (**A–C**) show clearly different alkali granite BGM compositions than Figure 4 (**D–F**) for syenite BGM.

The chondrite-normalized REE patterns for the BGM from the nepheline syenite pegmatite (Figure 4E) are mostly similar for all end-members and have a U-shaped form with a strongly fractionated pattern for LREE and an unfractionated pattern for HREE (La/Gd$_n$ = 7.2–26.1, average 13.4; Gd/Yb$_n$ = 0.28–0.67, average 0.42). However, fluorcalciobritholite contains lower La and Ce compared to Ce- and Y-dominant 'britholites' from the same pegmatite. La/Yb$_n$ in fluorcalciobritholite ranges from 3.7 to 7.6 (average 5.2) whilst in Ce- and Y- britholites the ratio has the values 4.6–8.8 (average 5.9). Fluorbritholite-(Ce) has no, or a slightly negative, Y anomaly (0.87–0.96) whilst all other britholites show a strong positive Y anomaly (1.0–1.9, average 1.45). Eu and Tb are below the detection limits.

## 4.3. BGM Substitution Schemes and the 'Calciobritholite'-'Apatite' Series

Compositional variation in BGM from the Keivy alkali granite-syenite complex is well represented by the coupled substitution $REE^{3+} + Si^{4+} = Ca^{2+} + P^{5+}$ (I) [40] (Figure 5). The lowest apatite (phosphorus content) components are in fluorbritholite-(Ce) from the apobasic metasomatite (0%) and in most britholite-(Y) from the granitic pegmatite (0–8%) and mineralized granite (0%), in fluorbritholite-(Y) from quartzolite-I (0–15%) and in britholite-(Ce) from trachytoid nepheline syenite (4–7%). BGM from quartzolite-II and the porphyritic nepheline syenite have moderate apatite components (10–25% and 7–15%, respectively). The highest apatite components in the BGM are found in fluorcalciobritholite and 'calciobritholite' from the nepheline syenite pegmatite and vary from 23 to 47%. According to the coupled heterovalent substitutions at the M and T sites in the series apatite-'calciobritholite'-britholite [25] the boundary between 'apatites' and 'calciobritholites' lies at Ca + P = 5. Thus, Keivy BGM fill in all the 'calciobritholite' compositions up to the apatite boundary (Figure 5A) and it is possible to expect a continual transition to REE-rich apatite. Nevertheless, the boundary between fluorcalciobritholite and fluorapatite from the nepheline syenite pegmatite in their intergrowths is always sharp (Figure 2). The same is found for coexisting britholite-(Y) and fluorapatite from Keivy mineralized granite (see Figure 7A in [38]) and for britholite-apatite intergrowths from Rodeo de los Molles [20]. This is consistent with the observation of [12] that fluorcalciobritholite forms a complete solid solution with 'true' REE-dominant britholite, but not with apatite.

Two other substitution schemes $Na^+ + REE^{3+} = 2Ca^{2+}$ (II) and $Th^{4+} + 2Si^{4+} = Ca^{2+} + 2P^{5+}$ (III) are insignificant for 'calciobritholites' and for a number of other BGM due to the negligible contents of Na and Th (see above). Only for 'britholites' from quartzolites and nepheline syenites with Na = 0.06–0.09 apfu is a slight shift below 1:1 line observed (Figure 5B), indicating an involvement of substitution scheme (II). The same pattern for single samples of the 'britholites' with Th = 0.06–0.1 apfu points to a minor involvement of scheme (III).

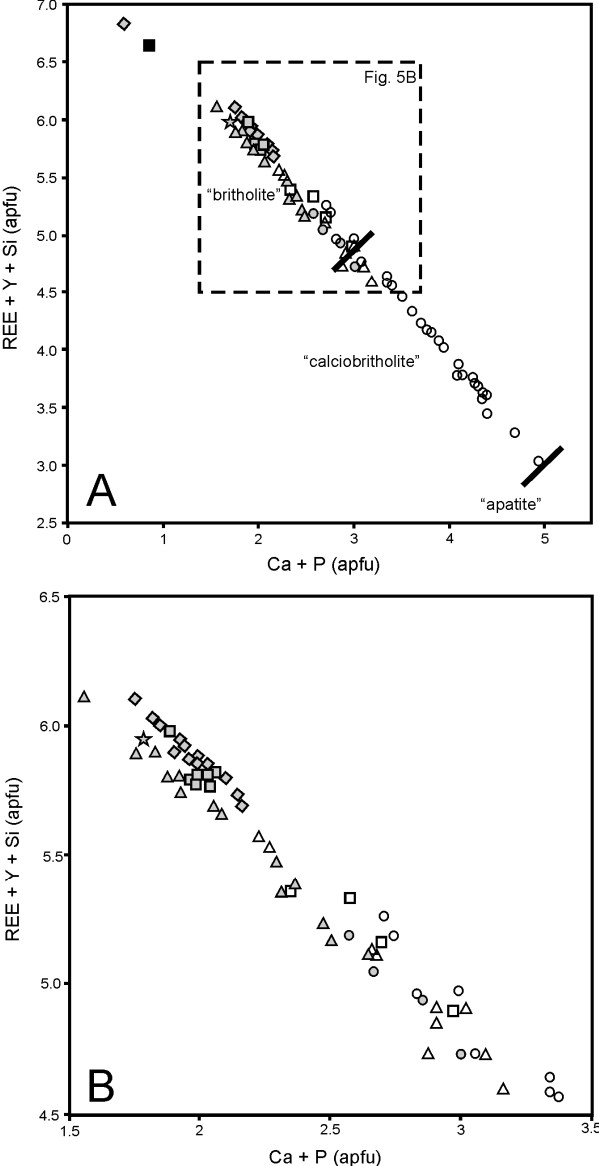

**Figure 5.** The dominant substitution scheme for the BGM from REE-rich lithologies related to the Keivy alkali granite-nepheline syenite complex (Figure 5 (**B**) is an enlarged dotted quadrant in the Figure 5(**A**)). The boundaries between 'britholite', 'calciobritholite', and 'apatite' are drawn according to the coupled heterovalent substitutions at the M and T sites and have Ca + P = 3 and 5, respectively [25]. Figure shows overlap of discrete populations. Symbols as in Figure 3.

## 5. Discussion

### 5.1. REE Enrichment Processes in Keivy Alkali Granite-Nepheline Syenite Complex

The strong enrichment in REE and high-field-strength elements (HFSE) characteristic of alkali granite-syenite complexes and their products can be related to two processes: enrichment through extended fractional crystallization of alkaline magmas and enrichment via late- and postmagmatic processes with the involvement of pegmatitic and hydrothermal fluids [22,41–47]. While not crucial, an enriched mantle source of the primary magma should be taken into account. Geochemical and Sr-Nd isotope studies of Keivy alkaline rocks have shown their parental magma's similarity to OIB magma [48]. High La/Yb$_n$, deep negative Eu anomalies, extremely high Fe-index (90–100%), low Ca are characteristics of the Keivy alkali granite and nepheline syenite and imply that the rocks

were formed by protracted fractional crystallization [27,30]. It is suggested that the main mechanism was the fractionation and removal of Ca-rich plagioclase causing the formation of associated coeval gabbro-anorthosite bodies and increases in alkalinity and lowering of Al in residual melts.

Undoubtedly, the dominant processes in the REE enrichment of the late- and postmagmatic products in the Keivy complex were fluid reworking and hydrothermal activity. Mineralized granites and syenites, quartzolites, and metasomatic rocks show the U-shaped chondrite-normalized REE pattern (Figure 6). Empirical and experimental data [49–53] have shown that a selective accumulation of HREE in late- and post magmatic rocks is promoted by high contents of alkalis (mainly sodium), fluorine, and $CO_2$ in metasomatizing or hydrothermal fluid/solutions. This agrees with the inferred scheme of evolution of the Keivy rocks: silicification (granites), albitization (syenites), and crystallization of hydrothermal minerals (fluorite, carbonates, zeolite group minerals) at the late- and post-magmatic (including pegmatitic) stages of massif formation have been previously established by petrographic study and zircon mineralogy [27,33,34]. The different degrees of autometasomatic and fluid reworking of the rocks and their significant role in the ore varieties is emphasized by the higher F contents and positive correlation between $Na_2O$ and HREE in the latter [21,27]. In addition, the mineralized granite and nepheline syenite have much higher U and Th contents and Th/U ratios, which are also well explained by late fluid reworking. The predominant accumulation of Th in the ores indicates a high alkali content and the significant prevalence of F over $CO_2$ in the ore-forming fluid [53]. Similar Th behavior with respect to U has been noted for many magmatogenic REE deposits that were subjected to intense reworking by hydrothermal alkaline-fluorine fluids [54–56]. It is suggested that partitioning of F into fluid is also promoted by a low Ca content in the melt.

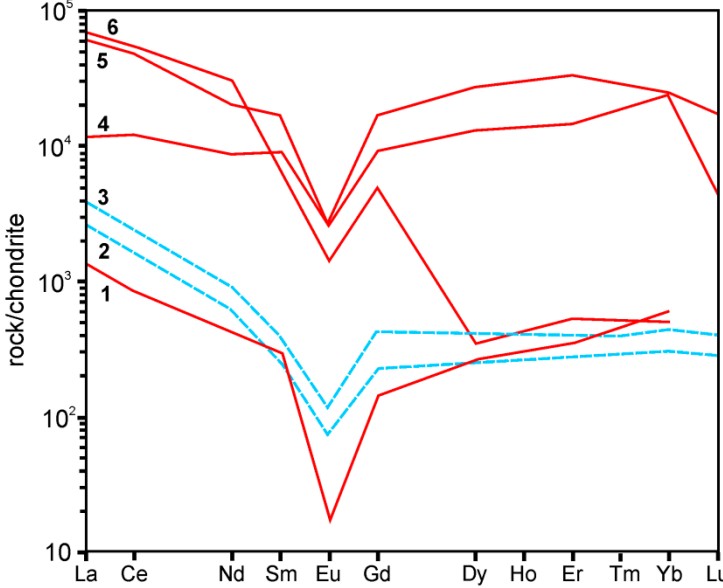

**Figure 6.** Chondrite-normalized REE patterns for REE-rich lithologies related to the Keivy alkali granite (red lines) and nepheline syenite (blue dashed lines): 1—mineralized granite; 2—mineralized porphyritic nepheline syenite; 3—mineralized trachytoid nepheline syenite; 4—quartzolite-II; 5—quartz-epidote (apobasic) metasomatite; 6—quartzolite-I. The data are from [21,27,32,55]. Normalizing factors from [39]. Figure shows that in some lithologies, i.e., in nepheline suenites, BGM are the main carriers of REE (cf. Figure 4D).

Thus, whole-rock geochemical and mineralogical data indicate that the composition of the hydrothermal fluids emanating from the Keivy alkaline magma was mainly alkaline and F-rich with changing proportions of $H_2O$ and $CO_2$. The role of other possible ligands (Cl, S) in the mobilization of the REE was unimportant due to their negligible concentrations in the parental alkali granite and nepheline syenite (Cl ≤ 0.02 wt %, S ≤ 0.1 wt %).

### 5.2. Fluid-Driven Diversity of BGM and Their Origin

As the BGM from the Keivy REE-rich lithologies have different F and REE concentrations, those components can be used to estimate the fluid composition. There is good experimental and theoretical evidence that all the REE form complexes with F but of different solubilities [42,57–59]. The important role of REE-F complexes in the formation of REE-rich deposits has been extensively studied by a number of authors [53,60–64]. For example, Y fluoride complexes are more stable than Dy fluoride complexes [65], thus the F enrichment in fluid will result in successive increases of Y/Dy ratio in the fluid until crystallization of F-rich minerals (normally fluorite, in some cases fluorapatite) occurs. The consumption of F destabilizes Y-F complexes, resulting in local crystallization of REE minerals with higher Y/Dy ratios. At that stage, Dy will tend to enter any REE minerals formed more efficiently than Y.

The highest Y/Dy ratios, ranging from 8 to 15, were observed in fluorcalciobritholite and 'calciobritholite' from the nepheline syenite pegmatite. Fluorbritholite-(Ce) from the same occurrence has lower Y/Dy (6–8). BGM from porphyritic nepheline syenite and micaceous metasomatite have Y/Dy ratios ranging from 5 to 7. The range 3–6 is determined for fluorbritholite-(Ce) from the trachytoid nepheline syenite. The ratio varies from 7 to 11 in fluorbritholite-(Y) from quartzolite-I and is about 7 in (fluor)britholite-(Y) from quartzolite-II and granitic pegmatite. The lowest Y/Dy ratio (about 2) was observed in fluorbritholite-(Ce) from the quartz-epidote metasomatite.

Thus, variations in the Y/Dy ratio may indicate changes in F concentrations in fluids not only between different lithologies but even in a single genetic series. Moreover, BGM have a general negative correlation between Y/Dy ratios and F concentrations in the minerals from a given genetic series (Figure 7). This phenomenon can be explained by two possible processes: (1) Ca from the fluid is mainly bonded in earlier crystallized fluorite and fluorapatite, after which the remaining fluorine in the fluid presumably enters the BGM, leading to successive F enrichment in minerals with temperature decrease; (2) fluorine activity may decrease due to the temperature drop or a sudden input of $H_2O$ into the fluid.

The difference in F contents of the crystallization media from the various lithologies can be explained by the overall geochemical conditions. Thus, the low F in the nepheline pegmatitic fluid could be due to its entry into abundant early-crystallized fluorapatite and fluorite. The highest F content in fluids operating in the quartz-epidote metasomatite is due to the entry of calcium into Ca-minerals (epidote, amphibole) and the release of F into the fluid. The high F contents in the nepheline syenite and quartzolite fluids can be explained by the overall high F contents in alkaline magmas and by the low Ca in it due to the evolved character of the nepheline syenite and alkali granite.

Another interesting observation is the overall tendency for Ce/Y enrichment in BGM crystallized from fluids with higher F content (Figure 7B, where Y/Dy is proxying for F). The trajectories are similar but are separate for the nepheline syenite and alkali granite BGM series that is obviously due to the different geochemical environments. It is well known that $YF^{2+}$ complexes are more stable in hydrothermal fluids than $CeF^{2+}$ (the $YF^{2+}$ stability constant is nearly five times as high as the corresponding Ce complex [66]). That is why the high F content in fluids favors the inferred precipitation of Ce-rich BGM. The different position of the trajectories for nepheline syenite and alkali granite BGM is determined by higher LREE/HREE ratio in the nepheline syenite system (see Figure 6).

Another informative geochemical indicator of fluid composition in the formation of postmagmatic REE minerals is $(La/Nd)_n$. Smith et al. [67] showed that the fractionation of La and Nd depends on the $CO_2$ content in the mineral-forming solutions which affects the $(La/Nd)_n$ ratio in REE minerals (>4 for $CO_2$-rich solution, <4 for $H_2O$-rich solution). Since the paragenetic environment studied in [67] differs from these of the Keivy occurrences, the conclusions given below are indirect.

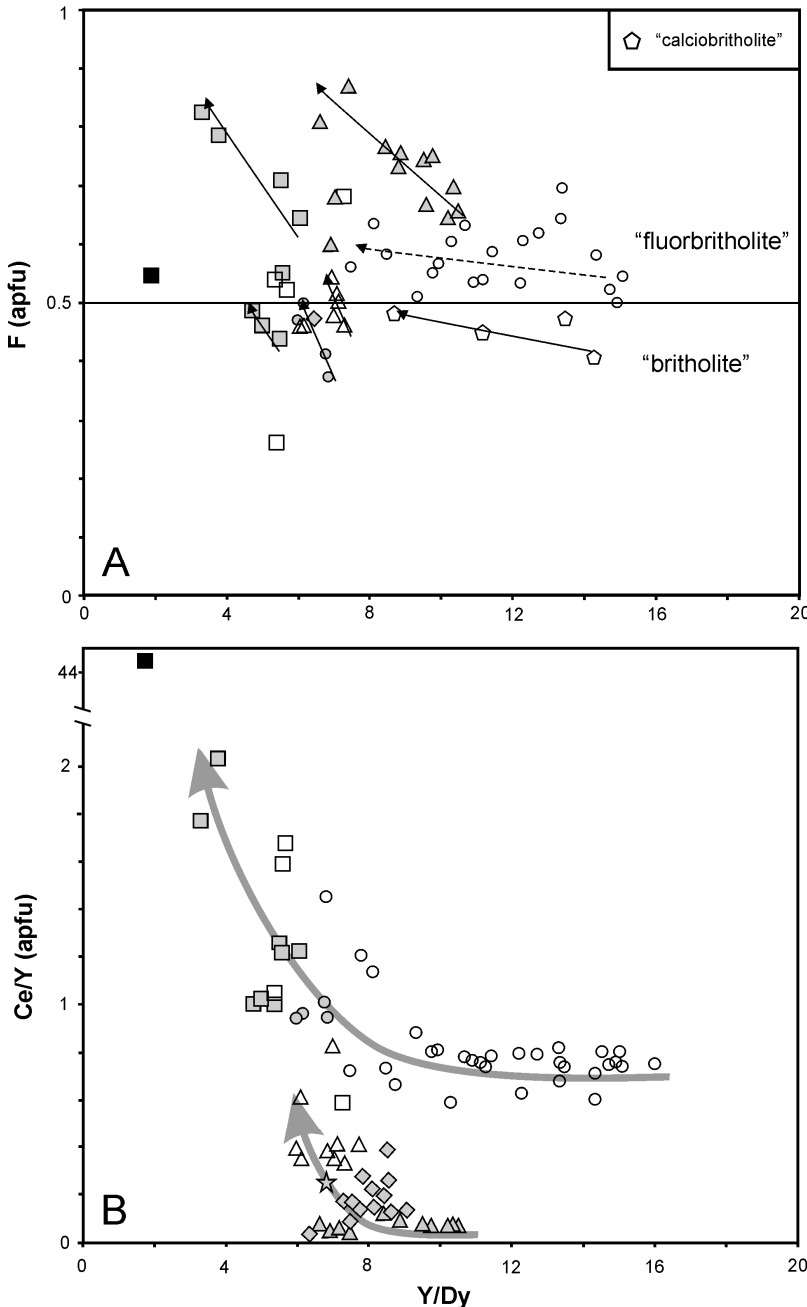

**Figure 7.** F vs. Y/Dy (**A**) in Keivy BGM, showing the negative correlation; Ce/Y vs. Y/Dy (**B**) in Keivy BGM, showing the increase in Ce component with lowering of Y/Dy for different genetic series of BGM: nepheline syenitic (upper trajectory) and alkali granitic (lower trajectory). Same symbols as in Figure 3.

The $(La/Nd)_n$ ranges from 4 to 16 in BGM from the nepheline syenite pegmatite, pointing to a relatively high $CO_2/H_2O$ in its fluid (Figure 8). BGM from the trachytoid nepheline syenite crystallized at moderate $CO_2$ activity ($(La/Nd)_n = 4–6$). The lowest $CO_2$ activity is suggested for fluids during BGM formation in the porphyritic nepheline syenite and micaceous metasomatite ($(La/Nd)_n = 2–4$) and for the alkali granite products ($(La/Nd)_n = \leq 2$ is documented in BGM from mineralized granite, pegmatite, quartzolites).

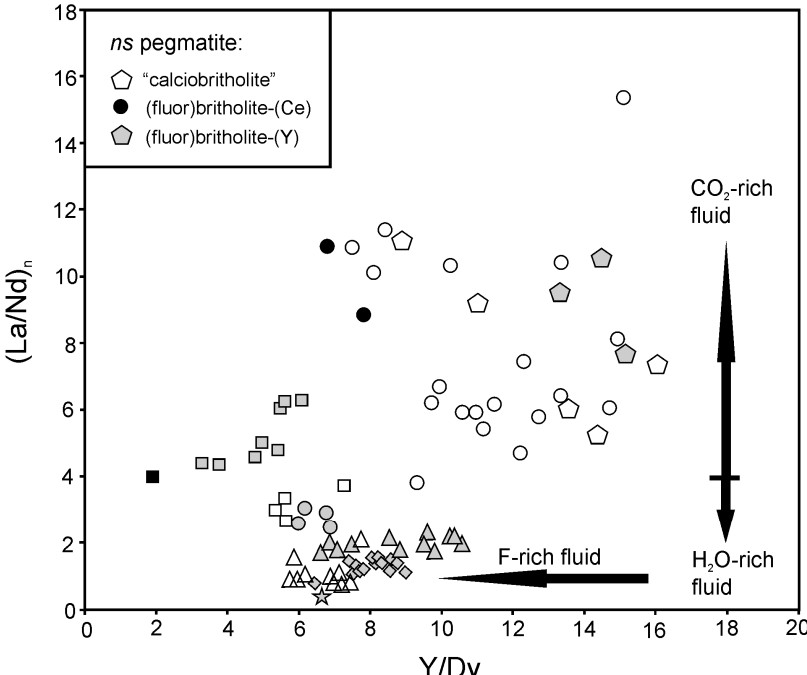

**Figure 8.** $(La/Nd)_n$ vs. Y/Dy plot for the BGM from REE-rich lithologies related to the Keivy alkali granite-nepheline syenite complex illustrating the different compositions of the fluids and how they resulted in the formation of different BGM end-members. Symbols and abbreviations as in Figure 3. Figure shows the variable fluid composition for different lithologies and changing fluid composition in a single system (i.e., in nepheline syenite pegmatite).

An extremely high scatter of $(La/Nd)_n$ and Y/Dy ratios in BGM from the nepheline syenite pegmatite can be explained by a changing fluid composition during the formation of the pegmatite. The lowest Y/Dy in the earliest fluorbritholite-(Ce) point to strong complexing of Y by F at that stage. Similar Y/Dy ratios are revealed in the pegmatitic fluorapatite at the same stage [26]. Furthermore, the precipitation of abundant fluorite due to F saturation of fluid during temperature decrease was followed by the formation of fluorbritholite-(Y). During crystallization of fluorcalciobritholite, the F activity goes on to decrease, which resulted in increasing values of the Y/Dy in the minerals. During this stage the sudden input of $CO_2$ or $H_2O$ into the fluid causes the crystallization of 'calciobritholite' with F < OH as rims on the fluorcalciobritholite.

A less significant change of F content and of $CO_2/H_2O$ ratio in the fluid is detected also for the nepheline syenite late- to postmagmatic system, and in the F content of fluid from the alkali granite-related quartzolite and pegmatite (Figure 8).

The lowest $CO_2/H_2O$ ratios in BGM from the granitic pegmatite and quartzolites can be explained by $CO_2$ saturation in the fluids, resulting in abundant precipitation of REE carbonates in these lithologies.

As the BGM from the alkali granitic pegmatite and quartzolite-I are very close in composition (i.e., Ce/Y ratio, REE distribution, (REE + Si)/(Ca + P), 'apatite' component; see Figures 3–5) and crystallized from fluids of similar composition (i.e., low $CO_2/H_2O$ ratio and moderate F content; Figures 7B and 8), it is possible to suggest the same formation conditions for these lithologies. This is consistent with their occurrence in the apical parts of granite bodies, while quartzolite-II is hosted by the country rocks.

## 6. Conclusions

1. REE enrichment of late- and postmagmatic lithologies of the Keivy alkali granite-nepheline syenite complex resulted mainly from protracted fractional crystallization of primary magma and from fluid-driven (both pegmatitic and hydrothermal) processes. The differences in evolution of two

systems (alkali granite and nepheline syenite) have led to different abundances and distributions of REE in 'britholite'.

2.　　Britholite group minerals in REE-rich lithologies related to the Keivy complex are fluorbritholite-(Y), britholite-(Y), fluorcalciobritholite, fluorbritholite-(Ce), and britholite-(Ce). The hypothetical mineral 'calciobritholite' has also been found. The dominant substitution scheme between the BGM species is $REE^{3+} + Si^{4+} = Ca^{2+} + P^{5+}$, showing no gap between britholite and 'calciobritholite' series even to the boundary with theoretical apatite.

3. Y-dominant species are confined mainly to alkali granite-related products (quartzolite, pegmatite, mineralized granite) and nepheline syenite pegmatite, while Ce-dominant minerals are characteristic of the mineralized nepheline syenite.

4. BGM are late- to post-magmatic minerals and crystallized from fluids with different proportions of F, $CO_2$, and $H_2O$. Fluorine contents, REE distribution, and the behavior of specific ratios (i.e., Y/Dy, Ce/Y, La/Nd) in BGM suggest a F-rich and low-$CO_2$ fluid for the granite-related pegmatites and quartzolites, F-rich, and $CO_2$-bearing fluid emanating from nepheline syenite to form economically important britholite ores, and a low-F and $CO_2$-rich environment for the nepheline syenite pegmatite. The increase in F contents in fluids of 'nepheline syenite' system resulted in the crystallization of BGM with higher Ce/Y ratios. To a lesser extent, this dependence also applies to the 'alkali granite' system.

5. The high $REE_2O_3$ content (36–65 wt %, average 50 wt %) in BGM and elevated $Y_2O_3$ content (17–35 wt %, average 27 wt %) in 'britholite-(Y)' from Keivy rare-metal rich lithologies make mineral a perspective REE raw material.

**Supplementary Materials:** The following is available online at http://www.mdpi.com/2075-163X/9/12/732/s1, Table S1: EPMA method summary, Table S2: Chemical composition and mineral formulae of BGM from Keivy alkaline rocks.

**Author Contributions:** Conceptualization, D.Z.; Methodology, D.Z., Y.S., and P.J.; Investigation, D.Z., L.L., R.M., and B.B.; Writing—Original Draft Preparation, D.Z.; Writing—Review and Editing, R.M. and B.B.; Visualization, D.Z., L.L., and Y.S.

**Funding:** This research was funded by Russian Government grant 0226-2019-0053.

**Acknowledgments:** We are grateful to Beata Marciniak-Maliszewska from the Inter-Institute Analytical Complex at the Institute of Geochemistry, Mineralogy, and Petrology, University of Warsaw, for technical assistance with microprobe analyses. Three anonymous referees are thanked for helpful comments.

**Conflicts of Interest:** The authors declare no conflict of interest.

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
