# Peer review of "Britholite Group Minerals from REE-Rich Lithologies of Keivy Alkali Granite—Nepheline Syenite Complex, Kola Peninsula, NW Russia"

_minerals, doi:10.3390/min9120732_

Round 1

Reviewer 1 Report

This manuscript contains a lot of good data which is worth publishing.  However major revisions are required: most importantly the conclusions are not supported by the data.  The data are fine, the conclusions are inappropriate. Primarily the authors have nicely quantified a fundamental igneous process in the fractional crystallization of alkaline igneous rocks. However they have in no way demonstrated that the alkali granites and the 40 my younger nepheline syenites are from the same source.

The authors must indicate the ages of all the igneous phases they sampled -- the ages in Zozulya et al., 2005, MUST be provided.

Note there are mislabelling issues -- when you open the file labelled supplementary table 2, the table is labelled 'Table S1'. Line 295 refers to Table S1 incorrectly -- it refers to the data in Table 1, within the manuscript.

Author Response

Dear Reviewer 1,

Thank you very much for comprehensive review and suggestions to the manuscript. They really help us to improve the paper.

Author's notes to overall/major comments:

Conclusion is rewritten and the part concerning of importance of fractional crystallization in evolution of both systems (alkali granite and nepheline syenite) is added.

The precise ages of Keivy igneous phases (granites, syenites, gabbro-anorthosites) are incorporated (lines 74, 170-172). The ages of all phases coincide within error limits, so we can conclude that they belong the same magmatic event. The age of nepheline syenite (2610 Ma) was obtained using the obsolete mass-spectrometer MI-1201T using the conventional TIMS method. So the caution should be involved for its interpretation. The other ages (granite, alkali feldspar syenite, gabbro-anorthosite) were obtained using a Finnigan-MAT 262 and are more reliable. Judging from geological setting it is believed that all phases are genetically associated.

We do not emphasize the source of rocks in this paper. It could be a single or more. Moreover the granite magma can be contaminated by crust. Of importance for the paper is that the granites and syenite are of the same magmatic event. The difference in magmatic and postmagmatic evolution of two systems, their geochemistry, reflected in difference of fluid composition and crystallized BGM. We tried to introduce this into text (see Conclusions, Figure captions and their interpretations).

Author's notes to minor comments:

Labels for tables have been amended throughout the text.

'Fluids metasomatically altered the country rocks, not granites' - corrected throughout the text.

'Dyke' is not correct really - changed for 'fissure-type intrusion'. Figure 6 is redrawn.

We think that the notice in Introduction about processing scheme for REE extraction from BGM is important as the many Minerals readers are interested in mineral processing.

Reviewer 2 Report

Some minor comments are attached as notes to the text.

Figure 1. The color for Imandra-Varzuga Belt is similar to the one for the amphibolites in the legend. I suggest to change one of them.

Sub-chapter 2. Geological Background and REE-rich Occurrences. The text is difficult to read and I suggest to be re-structured. The rock-types types could be described in the text but the geochemical and mineralogical data to be in a table inserted into text. In this way, the reader will be able to compare the info for all the rock-types and understand the relationships between them.

Table 1. Formulae on the basis of 8 total (M+T) cations. I suggest to add the state of oxidation for each element in the apfu.

Figure 5. I think this figure is not necessary, it does not add anything new or emphasize a previous explanation.

Lines 468 to 551. Difficult to read and understand the text. I suggest the authors to try to re-structure a bit the text, some sentences are not very clear.

Author Response

Dear Reviewer 2,

Many thanks for your comments and suggestions. They really improved the manuscript.

Author's notes to minor comments:

Figure 1: the color for Imandra-Varzuga belt has been changed;

Section '2. Geological background...' has been re-structured. We have inserted the new Table 1 with summary of rare-metal minerals and WR content of REE, Nb, Zr, Th, U for Keivy rocks;

The oxidation state is added for heterovalent elements Ce, Fe, Mn.

We believe that the Figure 5 is necessary as it clearly demonstrates the distribution of different BGM populations according to the main substitution scheme. The diagram is widely used in studies of apatite supergroup minerals.

Authors note to major comment:

We really spent a lot of time to discuss the possible ways of reorganization of Section '5.2. Fluid-driven Diversity...'. And we found no reason for it. It seems us the text follows through logically.

Reviewer 3 Report

The paper is well done.

Author Response

Dear Reviewer 3,

Thank you very much for helpful comments and corrections. Most of them have been incorporated into text.

Some notes:

The list of BGM is done with exception of the tritomite species. This is mentioned in the text.

Mieite-(Y) is excluded from the text, as the mineral has the similar chemical composition, but not approved by XPRD and/or SCXRD studies.

Round 2

Reviewer 1 Report

Paper presents useful information to guide REE exploration and mining targeting and beneficiation techniques with respect to interest in LREE or HREE. This is hinted at in the introduction, not specified and not necessary in the abstract, but would be good to put in the conclusions to explain the importance/utility of the results of this investigation. 

Minor formatting corrections needed in the new table 1. The manner in which the contents of the columns wrap impairs the presentation.

Typo line 76, 'suites' should be plural.

Use of the terms trends vs trajectories in fig. 7 is not critical -- either term is fine.  My point was that the trends/trajectories deserved more discussion in the main text. The discussion is OK, the source of the overlap in trends is addressed well, but the ultimate significance in terms of application of the results of this study is not clearly stated.   

Author Response

Dear Reviewer 1,

Thank you for useful comments and suggestion.

The note about the importance of "britholite" as a new possible REE raw material is included into Conclusions (see point 5). We reformatted the table 1 to be more readable. "Suite" in Line 76 is corrected. The different position of 'nepheline syenite' and 'alkali granite' trajectories is explained (Lines 523-525).

Sincerely,

Dmitry Zozulya